# Evaluating empiric antibiotic prescribing for hospitalized children in Mozambique through the introduction of a quarterly syndromic antibiogram: An implementation science protocol

**Darlenne B. Kenga**[1]☯*, **Jahit Sacarlal**[1]☯, **Mohsin Sidat**[2]☯, **Gustavo Amorim**[3]‡, **Harriett H. Myers**[4]‡, **Valéria Chicamba**[5]‡, **Kathryn T. Kampa**[4]‡, **Troy D. Moon**[4]☯

**1** Department of Microbiology, Faculty of Medicine, University Eduardo Mondlane, Maputo, Mozambique, **2** Department of Community Health, Faculty of Medicine, University Eduardo Mondlane, Maputo, Mozambique, **3** Department of Biostatistics, Vanderbilt University Medical Center, Nashville, Tennessee, United States of America, **4** Department of Tropical Medicine and Infectious Diseases, Tulane University School of Public Health and Tropical Medicine, New Orleans, LA, United States of America, **5** Pediatric Intensive Care Unit, Hospital Central de Maputo, Maputo, Mozambique

☯ These authors contributed equally to this work.
‡ GA, HHM, VC and KTK also contributed equally to this work.
* Darlene.bintikenga@gmail.com

## Abstract

Antimicrobials are the most frequently prescribed drug in pediatrics, with an estimated 37% of infants and 61% of hospitalized children having received them. Approximately 20–50% of prescriptions have been shown to be potentially unnecessary or inappropriate. The World Health Organization (WHO) estimates that the continued increase in antimicrobial resistance by the year 2050 will lead to the death of 10 million people per year. This paper describes a protocol to be used in a future study to evaluate the implementation of a quarterly syndromic antibiogram, aimed to improve the use of antibiotics for the treatment of pediatric bacterial infections at the Maputo Central Hospital, Mozambique. This study uses implementation science methods framed by the Dynamic Adaption Process (DAP) and RE-AIM conceptual frameworks to develop a multi-phase, mixed-methods evaluation utilizing qualitative and quantitative approaches. The pediatric inpatient services at HCM consist of approximately 18 physicians and 60 nurses. Additionally, the microbiology laboratory consists of eight laboratory technicians. We anticipate analyzing approximately 9,000 medical records. Qualitative methods include in-depth interviews with clinicians, laboratory technicians, and administrators to explore current knowledge and practices around antibiotic decision making, facilitators and barriers to intervention implementation, as well as acceptability and satisfaction with the intervention roll-out. Qualitative analysis will be performed with NVivo 12 software. Quantitative methods include extracting data from existing records from the pediatric ward of Hospital Central de Maputo (HCM) guided by the RE-AIM framework to explore intervention utilization and other factors influencing its implementation. Quantitative descriptive and inferential statistical analysis will be performed using R Studio statistical

**Data Availability Statement:** The deidentified data supporting this study are available at the Open Science Framework (OSF) repository and can be accessed via the following link: https://osf.io/r2kw4/.

**Funding:** This study was funded by the Fogarty International Center (Award Number D43 TW009745) of the National Institutes of Health. The content is solely the responsibility of the authors and does not necessarily represent the official views of the National Institutes of Health. The funders had no role in study design, data collection and analysis, decision to publish, or preparation of the manuscript.

**Competing interests:** The authors have declared that no competing interests exist.

software. The findings from this evaluation will be shared with hospital administrators and relevant national policymakers and may be used by the Ministry of Health in deciding to expand this approach to other hospitals. The expected results of this research include the development of standard operating guidelines for the creation, distribution, and use of a quarterly syndromic antibiogram for antibiotic decision making that is informed by local epidemiology. Findings from this study will be used to develop a larger multi-site trial in Mozambique.

## Introduction

Globally, antimicrobials are the most frequently prescribed drugs in pediatrics, with an estimated 37% of infants and 61% of hospitalized children receiving antibiotics [1–6]. Reports indicate that anywhere from 20 to 50% of the antibiotic prescriptions made are potentially unnecessary or inappropriate [7–10] and that many children receive broad-spectrum antibiotics for viral infections or receive courses of antibiotics that are prolonged for longer than necessary [10–12]. Excessive exposure to antibiotics increases the risk of serious side effects, increases healthcare costs, and contributes significantly to the global and local emergence of antimicrobial resistance (AMR) [13, 14]. It is currently estimated that the continued increase in AMR worldwide will result in 10 million deaths each year among all age groups [15].

The rate of AMR in developing countries is worrisome and increasing. In Africa, one contributor to this is the fact that many patients seek healthcare and medication advice outside of conventional health systems, such as with traditional healers or directly with local pharmacists, without seeing a physician first [16]. Furthermore, improper antibiotic prescribing among hospitalized patients, including errors in both administration and/or dose have been reported to range between 20 to 80% [10, 17–19].

The limited studies done to date in Mozambique are showing high rates of AMR. For example, one study reported rates of multidrug resistant *Escherichia coli* at 29% and *nontyphoidal Salmonella* at 54% [20]. Another study in hospitalized children found nearly 70% of *Staphylococcus aureus* were methicillin-resistant and roughly 50% of *Klebsiella* had extended spectrum beta lactamase (ESBL) production [21]. Finally, a study in Mozambique evaluating AMR patterns to the World Health Organization´s (WHO) list of essential antibiotics, found high rates of resistance to beta-lactam antibiotics (69.3%), gentamicin (70.6%), and cotrimoxazole (85.1%) [22].

The WHO emphasizes the important role of the microbiology laboratory in antimicrobial stewardship programs (ASP) [23]. In healthcare settings with limited microbiology capacity, healthcare professionals must choose an antibiotic regimen for their patients before laboratory results are available, which may result in harm to the patient if done incorrectly [24–26]. One important tool recommended for clinicians to use when making empiric antibiotic decisions is the antibiogram. ASPs, including the use of antibiograms, have been recognized globally as evidence-based interventions to combat the spread of AMR. Several studies [27–29] from different regions have demonstrated the positive impact of antibiograms on patient outcomes, including reduced mortality rates, improved appropriate antibiotic prescribing, and lowered healthcare costs. Furthermore, studies have shown that syndromic antibiograms, as compared to traditional antibiograms. increase the likelihood of effective empiric therapy for a specific infectious syndrome and can be further stratified based on hospital location [30].

When available, antibiograms offer a convenient snapshot of the pathogens identified by a given institution´s microbiology laboratory and their antibiotic susceptibility patterns [30].

*Traditional antibiograms* are the most easily available and report the proportion of pathogens identified and their susceptibility patterns within a given time period. However, they are limited in the data provided to a clinician, including a lack of syndrome specific recommendations; lack of information describing the distribution of organisms by a specific infection type; lack of recommendations for infections caused by more than one organism; and are typically generated retrospectively, such that susceptibility data may be outdated [30]. Over time, antibiograms have evolved in the type of information provided. For example, *Combination antibiograms* report on the probability that at least one drug in a multi-drug regimen covers a particular pathogen and provides a useful tool for clinicians to assess antimicrobial coverage. *Syndromic antibiograms* disaggregate pathogens and their antibiotic susceptibility patterns by infection type, such as by urinary tract infections (UTI), or by location within a facility, such as within the intensive care unit (ICU). Finally, a *Weighted-incidence syndromic combination antibiogram* (WISCA), combines all the above and further disaggregates reporting of results by patient characteristics such as age, gender, and comorbidities. Current studies serve as a valuable reference for the utility of antibiogram utilization, especially in situations like Mozambique where laboratory capacity is limited and there is a dearth of experience in their implementation. A quality, up to date antibiogram, that is disaggregated such that the clinician is informed of the most likely organism observed in each clinical infection syndrome as well as the likely resistance patterns of observed isolates, can be a key resource for clinicians when selecting empiric antibiotics [30–32].

In this study, based on current resources and capacity within our proposed study hospital, we aim to employ an implementation science approach to address key knowledge and research gaps towards the roll-out of a quarterly (once every three months) *Syndromic Antibiogram* within the pediatric in-patient service of the Hospital Central de Maputo (HCM) in Maputo City, Mozambique.

## Materials and methods

### Aims and objectives

The overall aim of this study is to evaluate the introduction and roll-out of a quarterly syndromic antibiogram, on the use of antibiotics for treatment of bacterial infections among pediatric patients admitted to HCM in Maputo, Mozambique, over a period of 12 months. The focus of this evaluation is on the implementation of the syndromic antibiogram. Secondarily, we will evaluate effectiveness based on clinical outcomes and the impact of antibiogram use on duration of hospital stay. As such, no individual patients from the pediatric ward will be enrolled in this study. Qualitative interviews with health care workers will take place over a maximum of a three-month period. No data collection or participant enrollment has taken place at the time of manuscript submission. The Specific objectives of our study include:

1. To explore the knowledge, attitudes, and practices of health professionals in the pediatric services and clinical laboratory of HCM regarding local antibiotic resistance trends and ASP in general.

2. To understand barriers and facilitators to the implementation of a syndromic antibiogram in pediatric patients suspected of bacterial infections and hospitalized at HCM.

3. To ascertain antibiotic resistance patterns for the organisms commonly seen by clinical syndrome.

4. To evaluate the implementation processes of the syndromic antibiogram and its impact on clinician choice of antibiotics for the empirical treatment of bacterial infections.

5. To measure the use and duration of antibiotics, clinical outcomes, and duration of hospital stay for patients managed with the syndromic antibiogram.

## Expected results of the research

1. To develop standard operating guidelines for the development of a quarterly syndromic antibiogram by the HCM microbiology lab, that is informed by local epidemiology and provides a workflow plan for its distribution and utilization by the pediatric clinical team.

2. To train pediatric clinicians on the interpretation of the syndromic antibiogram and on ASP strategies for the rational use of antibiotics.

3. Utilize data generated in this single-site, pilot study, to inform the development of a broader multi-site study by testing feasibility, refining methods, identifying potential issues, and building the foundation for a successful larger-scale investigation.

## Study setting

This study will take place in the pediatric ward of HCM. HCM is a 1500 bed, national reference center, located in Mozambique´s capital city of Maputo. It is the flagship teaching hospital within the Mozambican National Health System. The pediatric service has a capacity of 326 beds and its catchment area includes Maputo City and Maputo Province. Based on inpatient registries of the pediatric ward at HCM for the years 2022–2023, on average 9,500 children are hospitalized per year, of which approximately 95% (9,025) received antibiotics during their hospitalization.

The Microbiology laboratory at HCM conducts an array of common essential tests, including complete blood counts (CBC), biochemical assays, and urinalysis. In addition, the laboratory performs a variety of different point of care (POC) rapid tests (for example: HIV, malaria, and syphilis) and is equipped for molecular biology of priority infectious diseases such as HIV (DNA PCR and viral load) and Hepatitis B. Finally, the laboratory performs bacterial culture of blood, urine, stool, sputum, and cerebrospinal fluid, with associated antibiotic sensitivity testing. In total, the laboratory processes roughly 2.5 million samples per year, of which roughly 400,000 samples are from bacterial cultures. Current turn-around times for culture results are, on average, five days. This delay results in clinicians frequently making antibiotic choices empirically, rather than based on laboratory confirmed results.

## Study design and conceptual frameworks

We propose a mixed-method study, using an exploratory sequential design in which qualitative data results will inform utilization of the syndromic antibiogram (facilitators and barriers), followed by quantitative data results which will evaluate the antibiogram´s implementation. The study will employ two conceptual frameworks, the Dynamic Adaptation Process (DAP) and the Reach, Effectiveness, Adoption, Implementation, and Maintenance (RE-AIM) framework, and will be carried out in three implementation phases (**Fig 1**). The DAP framework was developed to provide the structure for an iterative process to guide, monitor, and evaluate the introduction of a new intervention into practice. DAP engages stakeholders at all levels to develop robust implementation strategies [33] and will guide the work of phases one, two, and three in our study. *Phase One* will occur over a maximum of three months, which will be based on pre-implementation steps that include 1) self-administered questionnaires that will be

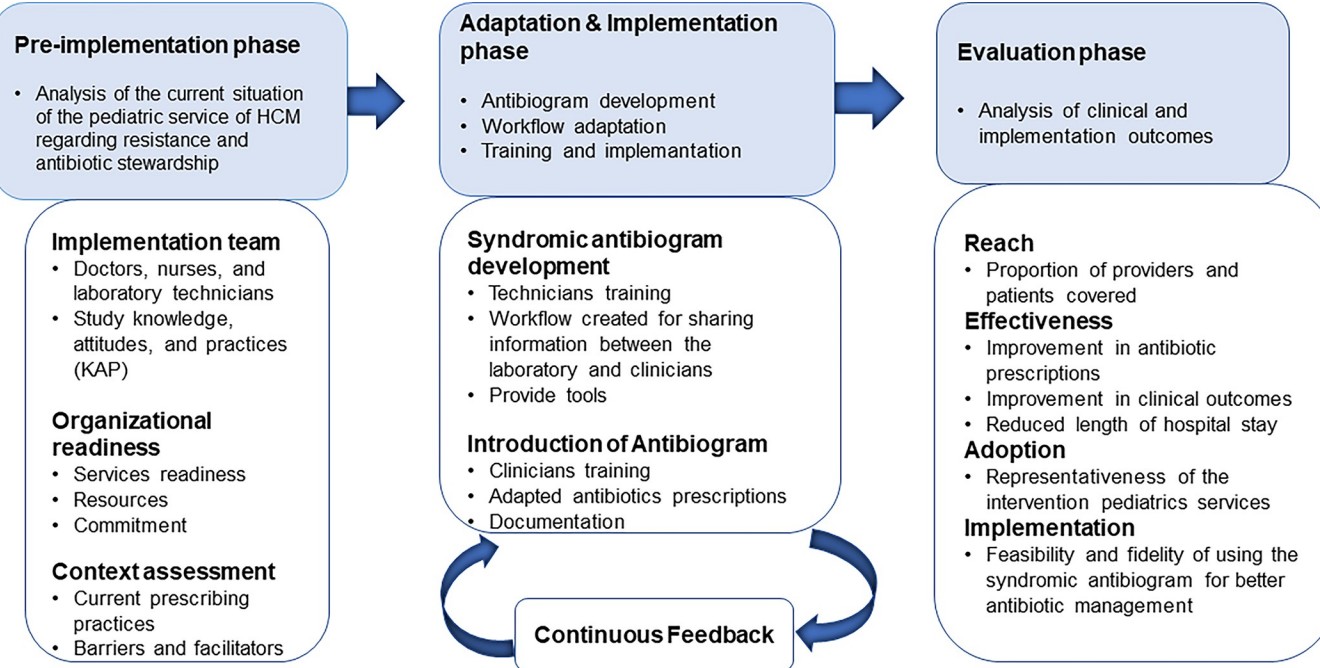

**Fig 1. Study interventions guided by the integrated Dynamic Adaptation Process and RE-AIM conceptual frameworks.**

distributed to physicians, nurses, pharmacists, and laboratory technicians at HCM to determine knowledge, attitudes, and practices related to AMR and ASP standards, followed by in-depth qualitative interviews with physicians, nurses and laboratory technicians to determine the barriers and facilitators to the implementation of a syndromic antibiogram among hospitalized pediatric patients at HCM; 2) training of laboratory technicians on the development of the syndromic antibiogram (S1 File); and 3) training of clinicians on its interpretation as well as a workflow analysis towards putting the syndromic antibiogram into practice (S2 File). The workflow analysis will assess antibiotic prescribing practices in the pediatric ward of HCM prior to implementation of the syndromic antibiogram. This will consist of analyzing standard operating procedures (SOP) and patient care decisions related to antibiotics during the flow from hospital admission triage to admission on the pediatric wards. We will map 1) what is happening at each step, 2) who is involved, and 3) inflection points where antibiotic decisions are made aiming to identify bottlenecks and inefficiencies. Through this assessment, areas for improvement will be pinpointed to streamline practices and enhance patient care. *Phase Two* will consist of an adaptation and implementation phase, informed by results in Phase One and will involve introduction of the syndromic antibiogram within the pediatric ward of HCM, followed by monitoring and evaluation of its effectiveness, acceptability, sustainability, and level of satisfaction of health professionals in its use. Phase two will also employ iterative learning cycles with updates made to the syndromic antibiogram every three months over a period of 12 months as needed, based on feedback provided from the clinicians and laboratory technicians for continuous improvement of activities. Finally, *Phase Three* will consist of a three-month post-implementation phase in which we will analyze implementation outcomes and processes as a function of the RE-AIM conceptual framework, in real time, in order to provide practical evidence-based key indicators of successful implementation of the syndromic antibiogram such as the feasibility of its use and the fidelity to the antibiograms defined roll-out procedures.

## Data collection activities

**Objective 1.** We will explore baseline knowledge, attitudes, and practices of HCM health professionals in the pediatric services and clinical laboratory, regarding local antibiotic resistance trends and ASP through data collected by means of a self-administered questionnaire. As of January 2024, the pediatric inpatient services at HCM consist of approximately 38 physicians and 60 nurses. Additionally, the microbiology lab consists of eight laboratory technicians. Recruitment and interviewing of all these available health professionals will be completed over the course of three months. The questionnaire will be adapted and piloted from a combination of questionnaires already available in the literature and will consist of multiple sections [34–37]. In the first section of the questionnaire will collect demographic, academic and professional data from the identified health professionals. The second section of the same tool will consist of questions assessing their knowledge and attitudes about antibiotics, AMR patterns in Mozambique and regionally, different ASP strategies, factors that contribute to AMR, sources of information to stay informed about AMS and ASP, and their confidence in prescribing antibiotics. Finally, the third section will explore current practices related to decision-making about antibiotic prescribing and the use of standard treatment guidelines, and advice and education provided to patients about antibiotic utilization.

**Objective 2.** Semi-structured qualitative interviews (approximately 18 total interviews) will be conducted with health professionals to identify barriers and facilitators to the implementation of a syndromic antibiogram in pediatric patients with a suspected bacterial infection and hospitalized at HCM. We will employ criterion based purposive sampling of the pediatric inpatient physicians and specialists (approximately 10 physicians) engaged in antibiotic prescribing to pediatric patients as well as laboratory technicians engaged in the conduct of antimicrobial susceptibility testing (approximately eight laboratory technicians). These cadre of health professionals are chosen based on their direct role in the provision of care to children admitted with a presumed bacterial infection. A semi-structured interview guide that was developed from the core frameworks will be used to prompt participants to discuss relevant themes and probe responses that are deemed relevant to better understanding the existing rationale in prescribing antibiotics to pediatric patients at HCM.

**Objective 3.** A syndromic antibiogram will be developed every three months for one year. The first antibiogram will be made just prior to study initiation. Then, in the last two-weeks of each three-month period a new antibiogram will be produced for use during the subsequent three-month period. Existing laboratory technicians will be trained and ultimately responsible for development of each antibiogram, with a goal that this is incorporated into the routine workflow of the lab. Microbiology data will be aggregated using WHONET software, already in use at HCM, producing susceptibility percentages for each microorganism identified in the laboratory. Identified isolates and their antibiotic susceptibility patterns characterized as resistant ("R"), intermediate ("I"), or sensitive ("S") will then be disaggregated by clinical syndrome (i.e., urinary tract infection, wound infection, etc.) and by location in the hospital from which the sample was collected (i.e., pediatric ward, intensive care unit, etc.). The designated study team including physician, pharmacist, and microbiologist will review these self-generated susceptibilities. Antibiotics susceptibility patterns included in the antibiogram will be restricted to those commonly available within the pediatric services of HCM.

**Objectives 4 and 5.** We will assess the _actual implementation outcomes/processes_ of the syndromic antibiogram, in real time, through quantitative and qualitative data collection in order to provide practical, actionable, information founded on key indicators of its successful implementation (Table 1). Per estimates of patients enrolled in 2022–2023 described above, we anticipate evaluating approximately 9,000 pediatric medical records over the one-year period

**Table 1. RE-AIM domains by data source and indicator/construct assessed.**

| Domain | Data Source | Indicators or Construct Assessed within each Domain |
|---|---|---|
| Reach | Medical Record review | Health Record Data collection to track antibiotic prescribing patterns among clinicians and to identify the number of patients whose antibiotic regimens were selected based on the antibiogram recommendations. |
| Effectiveness | Medical Record Review | Health Record Data collection to extract data on patient diagnoses, antibiotic prescriptions, laboratory tests ordered, and outcomes. To record information on antibiotic use, prescribed medications, patient outcomes, and length of hospital stay. |
| | Surveys and Questionnaires | Self-administered assessment of healthcare provider proficiency in generating and interpreting antibiograms (feasibility of antibiogram use) and to assess the satisfaction of healthcare providers in using the syndromic antibiograms in their daily routine (Likert Scale responses). |
| | Audits and Observations | Study staff administered assessments to monitor antibiotic prescribing practices (both antibiotic choice and/or decision to not give antibiotics), laboratory utilization, and adherence to antibiogram-guided therapy. |
| Adoption | Qualitative interviews | In-depth interview of healthcare administrators (chief clinician and nurse, Director of the Microbiology lab, hospital administrators) to determine the extent to which their willingness to adopt and integrate syndromic antibiogram recommendations into routine clinical practice. Assess factors influencing feasibility, such as perceived usefulness, ease of use, and compatibility with existing practices. |
| | Training Data | •Number of clinical and laboratory staff trained.<br>•Proportion of staff trained from entire personnel pool and proportion of those intended to be trained. |
| Implementation | Audits and Observations Medical Record Review Surveys and Questionnaires | Study staff administered assessments of:<br>•Number of quarters in which a syndromic antibiogram was created and shared with clinicians within the last two weeks of the quarter (Fidelity of the antibiogram SOP).<br>•Proportion of eligible patients whose antibiotics were chosen based on syndromic antibiogram recommendations.<br>•Median/mean number of days between admission and when a patient was placed on antibiotics based on the syndromic antibiogram recommendations.<br>•Proportion of clinician confidence in interpreting syndromic antibiogram<br>•Proportion of antibiotic prescriptions that align with syndromic antibiogram recommendations. |
| | Qualitative interviews | In-depth interviews with clinicians to gather information on their prescribing practices, awareness of antibiograms, and willingness to use them (acceptability). |

of data collection. The RE-AIM model was developed to increase the impact of health promotion interventions by assessing domains considered most relevant for real-world implementation, such as their ability to reach underserved populations and be adopted in diverse settings. Briefly, the _Reach_ domain refers to the percentage and characteristics of individuals who receive the intervention; _Effectiveness_ refers to the impact of the intervention, including expected and unanticipated outcomes; _Adoption_ concerns the percentage and representativeness of environments that adopt the intervention; _Implementation_ refers to the consistency and cost of delivering the intervention; and _Maintenance_ refers to the long-term sustainability at both the facility and the individual level [38]. Due to the pilot nature of this study, the Maintenance domain will not be evaluated.

## Data management

Semi-structured interviews will be conducted face-to-face in Portuguese (the national language of Mozambique) in a quiet and private location and audio recorded with participants' consent. The audio recordings will be transcribed verbatim and later translated to English. The transcript will be typed into a word processing document on a password protected computer. All potentially identifying proper names of people, places, and/or organizations will be redacted from the electronic transcripts. The transcripts will be sent electronically via a secure file exchange site to the Principal Investigators (PIs) on a regular basis for analysis and archiving. All transcripts will be password protected prior to sending. Audio-recordings will be destroyed after the completion of data analysis.

Pre-implementation phase quantitative data will be collected using an adapted KAP questionnaire and piloted from a combination of questionnaires already available in the literature, and responses will be anonymous [34–37]. The questionnaire will be divided into sections and will

include open, closed (yes or no), and Likert-type response options to questions (e.g., strongly agree, agree, neutral, disagree, or strongly disagree). The children's clinical and laboratory information will be collected using pre-designed forms, which will include only the variables of interest to the study. The interview guides, questionnaires, and forms utilized during the implementation and evaluation phases will be uploaded into REDCap, a secure online survey development and database management platform, on the tablet. All data collected from the health unit's records will be stored on a secure computer in the office of the Microbiology Department of the UEM Faculty of Medicine, accessible only by key study personnel. All reports shared with community members and health authorities will be aggregated and individual responses will not be identifiable.

## Analyses

**Quantitative analysis.** The quantitative data analysis will utilize the statistical software R, where descriptive analysis will be performed. Continuous variables such as age of patients and length of hospital stay will be summarized (mean or median, as well standard deviation and interquartile ranges). Percentage frequencies will be made of the following variables: gender, types of microorganisms identified, antimicrobial susceptibility, type of sample, class of antibiotics, use of antibiogram, patients covered, and professionals who adhered to the implementation (implementation fidelity).

The Chi-squared test or Fisher's exact test will be used to compare frequencies (gender, level of education, microorganisms, antimicrobial resistance, antimicrobials, source of information, and training in antimicrobial use). The chi-square test will also be used to check associations between the intervention (syndromic antibiogram) and the variables (level of education of the professional, availability of the antibiogram, and gender). The nonparametric Wilcoxon-Mann-Whitney test will be used to compare continuous variables.

Logistic regression will be performed to estimate the relationship between implementation of the syndromic antibiogram and factors impacting its utilization in clinical practice in the post-implementation period. Our principal outcome variable will be binary, representing whether or not clinicians use the syndromic antibiogram for patients with suspected bacterial infections. Additional factors involved in implementation will be included, such as workload on the hospital wards/laboratory, ability to interpret the antibiogram, and ease of use. The hypothesis being tested is the extent to which these factors influence the likelihood of clinicians utilizing the syndromic antibiogram in their routine practice.

The pre-post analysis will be used to compare mean KAP scores between professionals related to AMR and ASP standards before and after implementation of the syndromic antibiogram. Then we will employ Fisher´s least significant difference (LSD) and Sidak post-hoc tests to observe statistical significance. Similar approaches will be used to assess the effectiveness of the syndromic antibiogram in terms of clinical outcomes such as antibiotic use and duration of hospital stay, before and after its implementation. Linear or ordinal regressions will be used depending on whether the underlying assumptions are met. A p-value $<0.05$ will be considered statistically significant.

## Qualitative analysis

Immediately following each recorded interview, the audio file will be uploaded into a REDCap database. The audio files will be transcribed verbatim and a set of *a priori* codes derived from RE-AIM constructs will be applied by two researcher assistants. The coders will be blinded to the level of implementation adoption and fidelity to avoid bias. Using NVivo, a software program for qualitative and mixed-methods evaluations, we will prepopulate all deductive codes and guidelines to facilitate coder inter-reliability. We will categorize and organize the data

using a hybrid approach of qualitative methods of thematic analysis, allowing us to identify recurrent themes and explore interview responses. Deductive codes will be used to form main themes, while inductive subthemes will be data-driven and incorporated into the analysis as they emerge. A rigorous consensus-based coding process will be implemented with inter-coder reliability assessments conducted after each analyst has coded five interviews. Coding discrepancies will be discussed, and if consensus cannot be achieved, the study principal investigator will engage to arbitrate disputes.

The sample sizes for the qualitative data collection activities are based on purposive, non-probabilistic sampling where the size of the sample relies on the concept of saturation, or the point at which no new information or themes are observed in the data. Research has shown that saturation can occur within the first 12 interviews conducted in a relatively homogeneous group when the objective of the research is to understand common perceptions and experiences [39].

### Limitations

This study has several recognized limitations. First, as this is a single-institution study, our findings may not be representative of all health professionals in the country (physicians, nurses, pharmacists, and laboratory technicians). Next, our quantitative data collection relies on self-administered questionnaires and medical record reviews, potentially resulting in missing or incomplete data. To mitigate this, clear protocols and instructions for data collection will be developed to minimize missing or incomplete data. Quality control measures will be implemented during data entry and review to identify and address any discrepancies. More-over, multiple imputation or sensitivity analyses will be considered to assess the impact of missing data on study outcomes.

### Ethical considerations

Approval to conduct the study was obtained from the Institutional Committee of Bioethics in Health of the Faculty of Medicine/Maputo Central Hospital (CIBS FM& HCM) (2022/111). Administrative approval was also obtained from the participating hospital. Written informed consent will be obtained and documented for all interviews and surveys. Hospital and pediatric ward record reviews have been provided a waiver of informed consent from the IRBs as secondary data extraction.

### Conclusions

The emergence and spread of resistant pathogens is rapidly becoming a major threat to public health and a significant burden to patients worldwide, prolonging hospital stays and increasing healthcare costs and mortality [40–42]. This is a particularly urgent matter due to the decreasing number of antibiotics approved for use in children over the last few decades [43, 44]. Studies have shown that utilization of a syndromic antibiogram, employed as part of an institution´s antimicrobial stewardship program, can contribute to reducing antibiotic resistance [30, 31, 45]. Hence, the expected results of this research include the development of standard operating guidelines for the creation, distribution, and use of a quarterly syndromic antibiogram for antibiotic decision making that is informed by local epidemiology.

This study will evaluate the introduction and roll-out of a quarterly syndromic antibiogram, on the use of antibiotics for treatment of bacterial infections among pediatric patients admitted to HCM in Maputo, Mozambique. The quarterly syndromic antibiogram represents a valuable resource that can empower healthcare providers with real-time insights into local antimicrobial resistance patterns and guide evidence-based antibiotic prescribing practices at the point

of care. By leveraging the syndromic antibiogram as a learning tool, healthcare providers have the opportunity to enhance their clinical decision-making skills, optimize antibiotic therapy, and contribute to antimicrobial stewardship efforts. The implementation of the present study will positively contribute to the improvement of empiric antibiotic prescribing, as well as to reduce the length of patient hospital stay. Our results likely will have implications for other patient populations, other sectors of the hospital, as well as to other hospitals in the region/country. As a result, the findings from this evaluation will be disseminated in a number of ways, including presentations among prescribing health workers of other sectors of HCM, at research conferences, and in peer-reviewed journals. A contextualized syndromic antibiogram on the judicious use of antibiotics in Mozambican hospitals will also be published on the website of the Ministry of Health and the collaborating hospitals for easy access by other health care institutions in and outside Mozambique. In addition, the results will be shared at a dissemination forum that will bring together members of health management teams at both national and district levels, clinicians who prescribe antimicrobial drugs, researchers, members of the public and other key stakeholders.

## Supporting information

**S1 File. Training for lab technicians (Developing a syndromic antibiogram).** (PDF)

**S2 File. Training for clinicians (Interpreting a syndromic antibiogram).** (PDF)

## Acknowledgments

The authors are grateful to the Partnership for Research in Implementation Science Mozambique (PRISM) team at both University Eduardo Mondlane and Tulane University for the administrative support provided for this research. In addition, we would like to acknowledge the mentorship provided by Dr´s Sacarlal, Sidat and Moon for their support and guidance throughout the development of this protocol and preparation of this manuscript. Finally, the authors acknowledge Andrea Kenga for critical English editing and contributions to the readability of this manuscript.

## Author Contributions

**Conceptualization:** Darlenne B. Kenga, Jahit Sacarlal, Mohsin Sidat, Troy D. Moon.

**Methodology:** Darlenne B. Kenga, Gustavo Amorim, Valéria Chicamba.

**Project administration:** Harriett H. Myers, Kathryn T. Kampa.

**Resources:** Troy D. Moon.

**Supervision:** Jahit Sacarlal, Mohsin Sidat, Gustavo Amorim, Troy D. Moon.

**Writing – original draft:** Darlenne B. Kenga.

**Writing – review & editing:** Jahit Sacarlal, Mohsin Sidat, Gustavo Amorim, Harriett H. Myers, Valéria Chicamba, Kathryn T. Kampa, Troy D. Moon.

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
