## [Decision Letter · Decision Letter 0]

10 Jan 2024

PONE-D-23-33366Evaluating empiric antibiotic prescribing for hospitalized children in Mozambique through the introduction of a quarterly syndromic antibiogram: an implementation science protocolPLOS ONE

Dear Dr. Kenga,

Thank you for submitting your manuscript to PLOS ONE. After careful consideration, we feel that it has merit but does not fully meet PLOS ONE’s publication criteria as it currently stands. Therefore, we invite you to submit a revised version of the manuscript that addresses the points raised during the review process in addition to the editor comments below.

We look forward to receiving your revised manuscript.

Kind regards,

Obed Kwabena Offe Amponsah, PharmD, Ph.D.

Academic Editor

PLOS ONE

“Fogarty International Center, National Institute of Alcohol Abuse and Alcoholism of the National Institutes of Health under Award Number D43 TW009745. The content is solely the responsibility of the authors and does not necessarily represent the official views of the National Institutes of Health.”

Additional Editor Comments:

The authors present a good program of work for the proposed project to improve antibiotic use in the hospital in question. This has potential benefits beyond the hospital and as such invite the authors to revise this submission according to the reviewer comments. I hope this provides a comprehensive review that will improve the project implementation as well as patient outcomes subsequently. In addition to the reviewer comments, kindly provide context on the microbiology support the hospital/program has or will have (internally and or external to the hospital). This should include the tests typically carried out, number of samples processed annually (and by deduction, quarterly), lab turnaround times if available as well as culture positivity rates as most of this project is reliant on the microbiology lab for results to develop the antibiogram. Is there designated team member (s) assigned to developing the antibiogram quarterly and what is the proposed timeline for each antibiogram? Would each subsequent antibiogram be developed exactly at end of that quarter or at the beginning of the next quarter and how would this affect the proposed timelines? Additionally, a sustainability component is important for success beyond the project. Do the team members have protected time for these project activities, are the activities being carried out routinely or would there need to be additional commitment of time?

Reviewers' comments:

Reviewer's Responses to Questions

**Comments to the Author**

1. Does the manuscript provide a valid rationale for the proposed study, with clearly identified and justified research questions?

Reviewer #1: Yes

Reviewer #2: Partly

2. Is the protocol technically sound and planned in a manner that will lead to a meaningful outcome and allow testing the stated hypotheses?

Reviewer #1: Yes

Reviewer #2: Partly

3. Is the methodology feasible and described in sufficient detail to allow the work to be replicable?

Reviewer #1: Yes

Reviewer #2: No

4. Have the authors described where all data underlying the findings will be made available when the study is complete?

Reviewer #1: Yes

Reviewer #2: No

5. Is the manuscript presented in an intelligible fashion and written in standard English?

Reviewer #1: Yes

Reviewer #2: Yes

6. Review Comments to the Author

You may also provide optional suggestions and comments to authors that they might find helpful in planning their study.

Reviewer #1: Dear authors,

This is a good study, well done. See my comments below

1. The abstract should clearly state the number of participants/interviews that will be included or the number of prescriptions that will be reviewed.

2. The objectives seem to be too many, I hope they can be summarized or some of them combined.

3. Methods: Sample size estimation is missing in the protocol. At this stage, the authors should have and idea at the number of participants/prescriptions that will included in this study. It would be good to add the distribution of healthcare workers at the study site, i.e xx doctors, xx pharmacists, xx nurses, etc, but this may eventually change slightly at the time of data collection.

4. Limitations of the study: The authors should add some expected limitations to their study.

Reviewer #2: Thank you for the opportunity to review this manuscript. I also want to commend the authors for undertaking this important work in the service of improving outcomes for patients at Hospital Central de Maputo. Best of luck on rolling out this program.

This is a well-written manuscript that aims to fill a gap in the understudied area of implementation of interventions targeting antimicrobial resistance. Recognizing the overall research approach has been reviewed in the course of obtaining funding, my comments are intended to improve the manuscript's and study's impact. I am unable to evaluate details about the antibiogram as an intervention nor the contextual information about antimicrobial resistance, but I can comment on the implementation science methods and approach.

Overall comments:

1. The study focuses primarily on implementation of a quarterly syndromic antibiogram as an intervention. But considering the potential results of the knowledge, attitudes, and practice questionnaires and the pre-implementation phase, there may be an equally important need to de-implement current antibiotic prescribing practices. To what extent will your (pre-)implementation findings lead to use of de-implementation strategies and/or draw on de-implementation research and theory?

2. The role of using DAP to identify implementation strategies is underdeveloped. Do you have any a-priori ideas for which implementation strategies may be helpful and/or have other antimicrobial resistance-related interventions coupled with implementation strategies been specified in the literature? Do you plan to leverage tools like the Expert Recommendations for Implementing Change or processes like Implementation Mapping to select and adapt implementation strategies?

3. Align research objectives with the overall study aim and frameworks chosen. Given that the evidence-based intervention being implemented is the quarterly syndromic antibiogram:

a. specific objective 2 should be to elicit barriers/facilitators to implementing the antibiogram (not about rational use of antibiotics). Using a determinant framework can assist with preparing your interview guides for specific objective 2, analyzing your data, and selecting strategies.

b. specific objective 5 talks about measuring clinical outcomes and the Effectiveness implementation outcomes appear to be patient-related outcomes, yet the description says there will be no patient outcomes. Please clarify.

4. RE-AIM indicators are underspecified. While RE-AIM is an appropriate framework to use to specify what you should be measuring, the manuscript would benefit from further clarity. What are the actual Effectiveness measures of the syndromic antibiogram (and what's the intended effect size)? For Implementation, are there any measures of fidelity (this is alluded to in the figure) and what does consistent delivery of the intervention look like? Please be as specific as possible.

5. The manuscript describes using multiple methods, but not mixed methods. While the quantitative and qualitative strands seem appropriate and justified, there doesn't appear to be any blending of the two strands together. Could these data be brought together in a joint visual display? If this is mixed methods and not multiple methods, is this a sequential or a convergent design and which strand is the priority?

6. Introduction needs some context about implementation of antibiograms and/or other interventions targeting antimicrobial resistance. Have others done this and if so, what was their implementation experience? Are there results demonstrating the effectiveness of antibiograms as an evidence-based intervention on the outcomes of interest in this study? If antibiograms are a well-recognized evidence-based intervention and presumably there's a gap in their widespread access or use, then what are the factors that explain why they are not more readily available or used? Is this a quality gap, an access gap, both? Furthermore, it's unclear to a reader unfamiliar with antimicrobial resistance, how innovative this quarterly syndromic antibiogram is. Is the antibiogram innovative because access to this resource is limited in Mozambique? Or is it the specific type of antibiogram being used? Or is it more about how the information is being used to shape prescribing patterns? More clearly delineating how this evidence-based intervention compares to the standard of care and the attendant access/quality gap this program is attempting to address is thus needed.

7. Align figure 1 with description from the 'study design and conceptual frameworks' subsection and specific objectives. The way the figure is drawn appears to contradict the description of the phases and the attendant research questions. There are 3 phases in the description and the separate pre-implementation phase, yet there are only 3 total phases in the figure. Has the pre-implementation phase already been completed and if not, shouldn't that be phase 1? Assessing organizational readiness is mentioned in the figure, but not in the description. Similarly, the bullet points under the RE-AIM evaluation indicators don't align with the description (e.g. feasibility and fidelity are mentioned here but not in the description).

8. Specify the recruiting/sampling methods in greater detail.

- specific objective 1: which healthcare professionals will you recruit (all? stratified sample? criterion-based sample, and if so what are the criteria)?

- specific objective 2: what type of purposive sampling will you use? See: https://link.springer.com/article/10.1007/s10488-013-0528-y as a helpful resource.

- specific objective 4/5: are all of these quantitative or will you use any qualitative data to evaluate things like impact, consistency, or maintenance?

9. The study design is missing some key details.

a. Will there be any comparison relative to the quarterly syndromic antibiogram intervention?

b. How will the to-be-determined implementation strategies fit into the design and subsequent analysis? Implementation studies typically compare the relative effectiveness of 1 or more implementation strategies on a particular implementation outcome(s) given a known barrier to implementation.

c. What types of study designs and analytical approaches will be used for measuring impact? Is this a pre-post study or another type of quasi-experimental design comparing the antibiotic prescribing patterns and other effectiveness outcomes?

d. Please provide a reference for and specify the qualitative thematic analysis in greater detail. What specific theories will you be engaging with in this part of the analysis? How does your positionality influence the study? Please also mention your intended approach to achieving thematic and/or meaning saturation.

e. For the quantitative analyses, especially for the effectiveness measures, there should be some discussion about sample size and power calculations.

Line-by-line comments:

- line 55: what's the scope of this? Is this globally? In Mozambique?

- line 63: although the implications are important, what does 'reduction in economic productivity' mean? Is this about GDP? I would advise deleting this for brevity.

- line 64: I would replace 'developing countries' with a more precise choice. Does rising antimicrobial resistance fall along gradients of wealth, geography, healthcare access or inequities, or something else?

- line 76: I would recommend condensing this paragraph, especially the detailed description of different types of antibiograms. This can be replaced with more description about the access/quality gap this program attempts to bridge and more contextual information about implementing antibiograms from other settings.

- line 120: is there an alternate word choice for strategy? For implementation scientists, this will be confusing to read as the antibiogram is the evidence-based intervention, which is distinct from the implementation strategies used to help implement the antibiogram.

- line 145: do you have any further information about the training (appropriate for a supplemental file)?

- line 146: this workflow analysis seems very promising and fascinating as an implementation strategy. Can you provide any more information?

- line 191: in the context of an implementation science protocol, this section reads as unnecessary detail and can be deleted in the interest of brevity. However, if this is really critical for a reader who works in this field, then please do keep.

- line 201: sentence is confusingly written. It's also unclear what an 'implementation evaluation approach' is. It would be more clear to just say you are evaluating the implementation of a quarterly syndromic antibiogram. 'Evidence-based' is misleading in this sentence and could be replaced with actionable.

- line 212: although important, the data management section can be reduced in favor of more description of the recruitment and analytical processes.

- line 238: how will adherence be measured? This seems critical to both fidelity and effectiveness. Is there a gold standard measure?

- line 239: depending on the size of your samples, consider an exact test like Fisher's in lieu of a chi-square test.

- line 222: what are the questionnaires you are referring to?

- line 243: the analysis description is underdeveloped. Are you testing a hypothesis? What is the outcome variable of implementation being assessed? Is this a binary outcome of whether or not someone used an antibiogram? The description of the methods implies that the laboratories will be trained in antibiograms so is the question whether or not clinicians order an antibiogram? Or is this outcome based on some other binary measure

- line 245: where do these factors fit into the figure about the phases? Are there any hypotheses about these factors?

- line 246: what is LSD?

- line 255: what's the conceptual framework being used here? Is this being used as an a-priori codebook for deductive coding?

- line 262: are there any implications for adult prescribing? Or in other hospital settings besides the pediatric ward? What about the replicability of the intervention and strategies in other settings (in Mozambique)?

- line 272: check word choice of 'withal'

- line 274: this is the first mention that a goal of this study is to reduce hospital stays. Is this an effectiveness measure?

- line 274: I appreciate the diverse dissemination plan described here, but I would instead focus on describing how the quarterly syndromic antibiogram can be used as a learning tool for direct patient care in the context of this study. This is one of the most exciting aspects of this intervention, but it's currently underdeveloped.

- Figure: if possible, I recommend attaching any data collection tools or instruments (that are available) as supplemental files.

7. PLOS authors have the option to publish the peer review history of their article (what does this mean?). If published, this will include your full peer review and any attached files.

Reviewer #1: **Yes: **Dr Steward Mudenda

Reviewer #2: **Yes: **Scott Halliday

---

## [Author Response · Author response to Decision Letter 0]

15 Mar 2024

Please see uploaded reviewer response letter

---

## [Decision Letter · Decision Letter 1]

1 May 2024

PONE-D-23-33366R1Evaluating empiric antibiotic prescribing for hospitalized children in Mozambique through the introduction of a quarterly syndromic antibiogram: an implementation science protocolPLOS ONE

Dear Dr. Kenga,

Thank you for submitting your manuscript to PLOS ONE. After careful consideration, we feel that it has merit but does not fully meet PLOS ONE’s publication criteria as it currently stands. Therefore, we invite you to submit a revised version of the manuscript that addresses the points raised during the review process.

The current version presents a significant revision of the protocol as recommended by the reviewers and the editor. Kindly address the concerns in the reviewer comments as soon as possible to facilitate the processing of this protocol.

We look forward to receiving your revised manuscript.

Kind regards,

Obed Kwabena Offe Amponsah, PharmD, Ph.D.

Academic Editor

PLOS ONE

Journal Requirements:

Reviewers' comments:

Reviewer's Responses to Questions

**Comments to the Author**

1. Does the manuscript provide a valid rationale for the proposed study, with clearly identified and justified research questions?

Reviewer #2: Yes

2. Is the protocol technically sound and planned in a manner that will lead to a meaningful outcome and allow testing the stated hypotheses?

Reviewer #2: Yes

3. Is the methodology feasible and described in sufficient detail to allow the work to be replicable?

Reviewer #2: Yes

4. Have the authors described where all data underlying the findings will be made available when the study is complete?

Reviewer #2: No

5. Is the manuscript presented in an intelligible fashion and written in standard English?

Reviewer #2: Yes

6. Review Comments to the Author

You may also provide optional suggestions and comments to authors that they might find helpful in planning their study.

Reviewer #2: Thank you for the thorough revision, which has resulted in an improved manuscript. In particular, the supplemental files are a nice complement to the main text. Please see minor comments below:

1. Figure 1 - please proofread the contents of the figure:

- pediatric service o[f?] HCM...

- Doctors and nurses? Or doctors, nurses, and _?

- Space before (Tdl)

- Study KAP -> unclear if KAP is knowledge, attitudes, and practices or if KAP is a member of the implementation team

- Barriers and facilitators (standardize use of sentence or title case)

- Training and implementation phase

- Introduction of antibiogram (similar subject-verb agreement with Syndromic antibiogram development)

- Instead of 'Results framed [sp] by RE-AIM, rephrase with a bullet point of activities taken out in this phase to make consistent with the description of other phases (e.g., Analysis of clinical and implementation outcomes?)

- Improvement (sp) in antibiotics prescriptions

- Effectiveness not Efficacy (to align with in-text description and table 1)

- Improvement in clinical outcomes

- Maintenance: I understand the rationale not to measure this in a pilot study and this is appropriately described elsewhere. However, Figure 1 does include the Maintenance domain. I would remove this to align with the rest of the manuscript.

2. Table 1 - thank you, this is much improved.

- I would place the Reach>Qual Interviews to evaluate acceptability section under Implementation. While acceptability is an important precursor to reaching more patients, it's also an established implementation outcome. RE-AIM narrowly considers fidelity to be the primary component to implementation so it would also be ok to leave where it currently stands.

- Adoption: this could also include training data from the clinician and laboratory technician training (such as raw number of staff trained, the proportion of the entire site, and/or proportion of those intended to be trained). Should the purpose of these qual interviews be to assess factors related to adoption? Regardless, I concur that having the contextual qualitative data will be valuable here.

- Implementation: these all seem like important data pieces. Are they all related to fidelity? Or other constructs? Pursuant to the text in the 'Study design and conceptual frameworks' section, which of these will be used to measure feasibility?

- The Data Management section indicates the quant data will come from the KAP questionnaire. Is that the same as the 'surveys and questionnaires' mentioned in the RE-AIM table?

Quantitative analysis:

- Your logistic regression proposal looking at these factors and utilization looks fascinating.

- The interrupted time series description indicates you're comparing before and after KAP scores. If there are only 2 time points, then this would be a pre-post analysis and not an interrupted time series. Or are there multiple timepoints?

Qualitative analysis:

- I don't quite follow the sentence on how deductive codes will be used to formulate main themes. In a thematic analysis, the themes typically result from relationships between codes (regardless of inductive or deductive coding).

- I appreciate the section about Diffusion of Innovations Theory and recognize this may be in response to one of my suggestions about how your qualitative analysis will engage with theory. However, with the clarification that deductive codes will be derived from the RE-AIM framework and the expanded description in Table 1, this now is a bit confusing. I'd now consider removing this paragraph.

Limitations:

- I disagree that DAP and RE-AIM lack standardization and that this is a limitation. RE-AIM in particular has extensive resources, standard definitions, checklists, and a guidebook for use on re-aim.org. Regardless, any heterogeneity in their use across different settings and contexts is to be expected. This is even a strength of an implementation evaluation given that you want to understand and harness this context towards identifying implementation strategies that are locally responsive.

- I also disagree that these RE-AIM measures may be subjective or that this is a limitation. Using a framework like RE-AIM gives you an assessment of domains known to be associated with high quality implementation. Adoption, reach, fidelity, acceptability, and feasibility are important intermediates to improving effectiveness outcomes. Furthermore, the rigor of your methods -- e.g., by establishing inter-coder reliability or using repeated measures before and after implementation -- may attenuate this concern. Lastly, even if participants do impart their subjectivity into the data, this can be helpful towards improving implementation.

-I'd strongly consider removing these limitations and instead focus on the limitations of your data collection/analysis methods.

7. PLOS authors have the option to publish the peer review history of their article (what does this mean?). If published, this will include your full peer review and any attached files.

Reviewer #2: **Yes: **Scott Halliday

---

## [Author Response · Author response to Decision Letter 1]

3 May 2024

Please see uploaded response letter

---

## [Decision Letter · Decision Letter 2]

20 Jun 2024

Evaluating empiric antibiotic prescribing for hospitalized children in Mozambique through the introduction of a quarterly syndromic antibiogram: an implementation science protocol

PONE-D-23-33366R2

Dear Dr. Darlene Kenga,

We’re pleased to inform you that your manuscript has been judged scientifically suitable for publication and will be formally accepted for publication once it meets all outstanding technical requirements.

Kind regards,

Obed Kwabena Offe Amponsah, PharmD, Ph.D.

Academic Editor

PLOS ONE

Additional Editor Comments (optional):

Reviewers' comments:

Reviewer's Responses to Questions

**Comments to the Author**

1. Does the manuscript provide a valid rationale for the proposed study, with clearly identified and justified research questions?

Reviewer #2: Yes

2. Is the protocol technically sound and planned in a manner that will lead to a meaningful outcome and allow testing the stated hypotheses?

Reviewer #2: Yes

3. Is the methodology feasible and described in sufficient detail to allow the work to be replicable?

Reviewer #2: Yes

4. Have the authors described where all data underlying the findings will be made available when the study is complete?

Reviewer #2: Yes

5. Is the manuscript presented in an intelligible fashion and written in standard English?

Reviewer #2: Yes

6. Review Comments to the Author

You may also provide optional suggestions and comments to authors that they might find helpful in planning their study.

Reviewer #2: Thank you for the revisions, which have resulted in an approved manuscript. Best of luck carrying out this important work and I look forward to seeing the results.

7. PLOS authors have the option to publish the peer review history of their article (what does this mean?). If published, this will include your full peer review and any attached files.

Reviewer #2: **Yes: **Scott Halliday

---

## [Editor Report · Acceptance letter]

2 Jul 2024

PONE-D-23-33366R2 

PLOS ONE

Dear Dr. Kenga, 

I'm pleased to inform you that your manuscript has been deemed suitable for publication in PLOS ONE. Congratulations! Your manuscript is now being handed over to our production team.

Kind regards, 

on behalf of

Dr. Obed Kwabena Offe Amponsah 

Academic Editor

PLOS ONE